# A Basic Study on the Performance Evaluation of a Movable Light Shelf with a Rolling Reflector That Can Change Reflectivity to Improve the Visual Environment

**DOI:** 10.3390/ijerph17228338

**Published:** 2020-11-11

**Authors:** Heangwoo Lee

**Affiliations:** College of Design, Sangmyung University, Cheonan-si 03012, Chungcheongnam-do, Korea; 2hw@smu.ac.kr

**Keywords:** light shelf, reflectivity, rolling type, visual comfort, performance evaluation

## Abstract

In recent years, the need for comfortable visual environments in indoor spaces has increased energy use in buildings. There have been diverse studies on using a light shelf to solve this problem. It is an effective system that allows external natural light deep into indoor spaces through reflection. However, prior studies have used light shelves with a fixed reflectivity, and there are few studies on improving the visual environment through light shelf control. Therefore, this study proposes a movable light shelf with a rolling reflector that can change the reflectivity. To achieve these objectives, we conducted a performance evaluation of the system’s ability to save energy and improve the visual environment. This study built a real scale testbed and conducted a performance evaluation by deriving values for lighting energy consumption, uniformity, and luminance contrast depending on the light shelf variables. We conclude that (1) the light shelf system achieved an energy savings of 13.6% and 5.7%, respectively, compared to a fixed type light shelf, whose reflectivity cannot be changed, and a traditional movable light shelf; (2) in terms of improving the visual environment, results suggest that the visual environment could be improved using a light shelf by deriving light shelf variables that disturb indoor uniformity; and (3) the results verified glare generation conditions by deriving luminance contrast caused by the variables of light shelf angle and its reflectivity.

## 1. Introduction

Creating a comfortable environment in an indoor space requires controlling various equipment such as lighting, ventilation, air conditioning, and heating equipment, rapidly increasing building energy consumption [1,2,3,4,5,6,7]. According to the “2018 Renewable Energy Data Book” released by the Department of Energy in 2020 [8], the building sector accounts for 39.8% of total energy use. Thus, the importance of studies on building energy reduction and relevant technology development is gradually increasing. In the building sector, as much as 17% of the sector’s total energy consumption is for lighting, according to “Electricity use in U.S. commercial buildings by major end uses”, which was released by U.S. Energy Information Administration in 2012 [9]. The number of studies that examine natural lighting systems as a solution to this problem is rapidly increasing [10,11,12]. There have been diverse studies on the light shelf, an effective system that inflows external natural light deep into indoor spaces through reflection [13,14,15,16,17,18,19,20,21,22,23,24,25,26,27]. While it is advantageous to increase the light shelf reflector’s reflectivity [25,26], it can give the indoor space occupants an uncomfortable visual environment [27]. Nevertheless, prior studies on light shelves evaluated lighting performance, such as energy savings, by fixing the reflectivity of a reflector to a specific value [13,14,15,16,17,18,19,20,21,22,23,24,25,26,27]. Because the majority of the previous studies on light shelves focused on lighting performance only, there has been little information about whether light shelves provide an appropriate visual environment [28].

Building on this, the current study aims to develop a movable light shelf with a rolling reflector that can change its reflectivity, and further evaluate the reflector performance in consideration of lighting energy saving and glare of indoor spaces to verify its effectiveness.

### 1.1. Experiment Equipment and Method

As shown in Figure 1, a light shelf is a natural lighting system installed on a window that introduces natural light from the outside by reflecting the light shelf reflector and the ceiling surface of the indoor space, reducing energy used for lighting [29]. A light shelf can also create a comfortable visual environment by addressing the unbalanced distribution of illumination in indoor space by blocking excess natural light coming through a window. Light shelf variables that determine its performance include width, angle, height, and reflectivity. The light shelf may be an internal or external type depending on the installation location [30].

As shown in Table 1, there have been diverse studies on improving lighting performance and the indoor visual environment. As shown in Figure 2, many studies varied the light shelf’s height and angle to improve lighting performance [16,18,20,21,22,23,24,25,26,27]. However, changing reflectivity was difficult because it required changing the light shelf’s angle or height. Therefore, previous studies evaluated shelf performance with fixed reflectivity. Some studies [16,21,25] considered a light shelf’s specific reflectivity when designing and applying it. If light shelf reflectivity is high, the lighting performance improves because more light enters the room through reflection. However, in some cases, this can cause excessive glare for occupants [31]. Meanwhile, if a light shelf’s reflectivity is low, the amount of natural light entering through the light shelf decreases, and the lighting performance deteriorates. In this case, the glare issue can be improved. Therefore, a light shelf’s reflectivity is an important variable that can reduce lighting energy in an indoor space and improve its visual environment. Therefore, the development of a light shelf capable of changing its reflectivity is considered in the current analysis, which differentiates this research from prior studies.

### 1.2. Review of Conditions for a Comfortable Visual Environment

Achieving a comfortable visual environment calls for various requirements. However, in relation to the light shelves, we limited the requirements to three conditions, i.e., maintaining adequate indoor illumination, balancing indoor illumination, and solving the glare problem [32,33,34,35,36]. The details are as follows.

First, what is considered appropriate illumination of an indoor space depends on the application characteristics and conditions. It also varies by country. As shown in Table 2, the current study thus reviewed illumination standards for indoor spaces proposed in the U.S., Japan, and Korea. There are minimum, standard, and maximum values depending on the application type. Based on the illumination standards proposed in the U.S., Japan, and Korea, the most common value of 500 lx was used in the current study as an appropriate level of illumination for an indoor space for our performance evaluation. We excluded 600 lx because it was the maximum allowable range of indoor illumination standards in Japan and Korea. Second, a severely imbalanced illumination of an indoor space causes visual discomfort and decreases optical task efficiency [37]. Therefore, illumination uniformity was calculated, which represents illumination uniformity in an indoor space. The uniformity calculation is derived from the minimum to average illumination ratio or the minimum to maximum illumination ratio. Third, a glare means the visual discomfort caused by excessive luminance or luminance contrast, so it is desirable to reduce the value of luminance contrast in an indoor space. In particular, a glare due to luminance contrast usually occurs under the following conditions [38]. An unpleasant glare occurs due to luminance contrast when the luminance value of a specific object is 10 times or more than the average luminance in an observer’s field of view. When the illumination value exceeds 25,000 cd/m^2^ due to direct sun exposure or a light source in an observer’s field of view, the excessive amount of light causes glare. In this case, a disability glare occurs regardless of the luminance contrast.

## 2. Method

### 2.1. Proposal of a Light Shelf with a Rolling Reflector With Varied Reflectivity

As shown in Figure 3, a light shelf with a rolling reflector that can change its reflectivity was proposed.

First, as shown in Figure 3, the proposed light shelf has connected reflective films with 70%, 85%, and 97% reflectivity, respectively. The film can be rolled by two axes to change reflectivity. The reflectivity values of 70% and 85% were chosen based on prior studies [13,14,18,22,23,24], and the reflectivity value of 97% was the reflective film manufacturer’s specification [39]. This is the result of an adjustment because lighting performance increases with reflectivity.

Second, two motors are used for the light shelf, i.e., a motor to control the light shelf’s angle and a motor used for rolling to change reflectivity. As a result, the light shelf is thicker than a typical light shelf because it has two motors and an additional apparatus that helps the reflective films rolling. Also, the light shelf’s width is higher than that of a typical light shelf due to its driving part. In other words, a light shelf with a rolling reflector that can change reflectivity must be wider to achieve the same reflective area as a typical light shelf. This proposed light shelf’s thickness and width can be changed depending on the reflective surface width.

Third, the control of the proposed light shelf with a rolling reflector that can change its reflectivity and its appropriate variables were based on the following procedure. Before an occupant enters the room, the light shelf is configured at an angle of −10° and a reflectivity of 70%. When the occupant enters, the angle is increased by 10° from its initial value. At a specific angle, its reflectivity is changed to 70%, 85%, and 97% to collect indoor illumination information. Based on the collected information, the light shelf automatically returns to an angle that can increase the visual environment’s comfort. The current study aims to improve indoor illuminance uniformity and luminance contrast for the optimal indoor visual environment while giving the highest priority to lighting energy saving.

### 2.2. Configuration of Performance Evaluation Environment

In the current investigation, a real-scale testbed was constructed, as shown in Table 3 and Figure 4, to evaluate the movable light shelf with a rolling reflector that can change reflectivity. The details are as follows.

First, the dimensions of the testbed constructed for the performance evaluation is 4.9 m (W) × 6.6 m (D) × 2.5 m (H), and the reflectivity of the indoor space is set to 86% for the ceiling, 46% for the wall and 25% for the floor. The skylight on which the light shelf is installed is 1.9 m (W) × 1.7 m (H) with pair glass applied, and the window transmissivity is 80%. 

Second, an external environment for performance evaluation was developed by constructing an artificial solar irradiation apparatus, as shown in Figure 4; Figure 5, outside the installed window. Artificial solar irradiation apparatus used in the current study can simulate the sun’s brightness and altitude by controlling the light amount, height, and angle of a light source. In particular, the artificial solar irradiation apparatus is an artificial solar irradiation apparatus of ASTM (American Society for Testing and Materials) E927-85 standard [40] A grade, so it is possible to derive valid experimental results. However, the mechanical characteristics of the artificial solar irradiation apparatus prevented it from simulating the sun’s azimuth angle. Its azimuth angle was limited to the south direction. When creating an external environment using the artificial solar irradiation system, the altitude and external illuminance were set for the summer, middle season, and winter, as shown in Table 4, based on the related studies performed in Seoul, Korea [26]. 

Third, in the current study, LED-type lights, which support nine-step dimming control, including lighting OFF, were installed at four locations to verify the light shelf’s lighting energy saving performance. Figure 6 shows the light distribution curve and conical illuminance according to the LED lighting’s nine-step dimming control status. The location of LED-type lights was determined based on the IES (Illuminating Engineering Society) four-point method [33]. Further, an energy monitoring system was installed to measure lighting energy consumption, and its measurement error rate is within 2%. 

Fourth, eight indoor illuminance sensors were installed to measure lighting control and indoor illumination distribution. The illumination sensors’ location was set to 4.4 m from the skylight, as shown in Figure 5, based on the study [41], which reported that 4.4 m is most suitable to measure the most representative illuminance value of an indoor space. The illumination sensor was located 0.75 m from the floor surface because it was the height of a typical work surface.

### 2.3. Performance Evaluation Method

In the current study, a performance evaluation of the lighting energy saving, uniformity, and glare of a movable light shelf with a rolling reflector that can change its reflectivity was conducted based on the following method.

First, three different cases for the performance evaluation were developed to verify the effectiveness of a light shelf whose reflectivity can be changed (see Table 5). The height of the light shelf for performance evaluation was set to 1.8 m from the floor of indoor space by considering relevant studies [28] and occupants’ eye level. Case 1 is a type with a fixed light shelf angle of 0°, and Case 2 is a movable type with a light shelf angle from −10° to 30° with 10° steps. From these two cases, lighting performance and glare were compared. The reflectivity of the light shelves used in Case 1 and Case 2 was set to 85%. The light shelf with a rolling reflector that can change its reflectivity is used in Case 3. The light shelf with a rolling reflector that can change its reflectivity (Case 3) is wider than that of typical light shelves used in Case 1 and Case 2 because it incorporates a rolling-type driving part to change its reflectivity. However, the actual reflective area of Case 3 is the same as for Case 1 and Case 2.

Second, the lighting control level and lighting energy consumption to maintain 500 lx, the appropriate illumination of an indoor space, were derived by analyzing illumination distribution in an indoor space for the three cases defined above. Lighting control is based on the following procedure. First of all, the lights 1, 2, 3, and 4 were linked to the illumination sensors 2, 4, 7, 9, respectively, for lighting dimming control. It enabled lighting control by measuring the values of illumination sensors 2, 4, 7, and 9. A lighting control system, based on illumination sensor information, was built in cooperation with Samsung SDS in Seoul, Korea. The lighting dimming control was configured such that there was no separate lighting control if all the measured values of illumination sensors 2, 4, 7, 9 are 500 lx or more, and there was lighting dimming control only when there was a measured value less than 500 lx. In this case, the dimming step increased step-by-step for a light linked to the sensor with a minimum value among the measured values of illumination sensors 2, 4, 7, 9. The lighting dimming control terminated when all the illumination sensors met 500 lx during this process. However, suppose 500 lx could not be measured at the sensor with a minimum illumination value, even after the eight-step dimming control. In that case, it was necessary to increase the dimming level for the light closest to the sensor and monitor illumination sensor information again. For example, suppose 300 lx was the minimum illumination value measured among illuminance sensors 2, 4, 7, and 9, and it was measured at illumination sensor 2. Then it was necessary to start dimming control for light 1, which was connected to illumination sensor 2, and check if all the measured values of the indoor illumination sensors met 500 lx while increasing the dimming level of light 1. If the indoor illumination sensor did not meet 500 lx even after dimming step 8 of light 1, the dimming step for light 3 was increased, which was closest to light 1, and we checked if the indoor measured values illumination sensors met 500 lx. This process was repeated, and the lighting dimming control terminated when all the indoor illuminance sensors met 500 lx. The lighting energy consumption was derived based on the lighting dimming control steps at this time. The lighting energy consumption was derived based on one hour in the south direction, and day 15, day 30, and day 15 were set for the summer, middle season, and winter, respectively; these are reflected in the performance evaluation.

Third, this study derived an index for illuminance uniformity in an indoor space and applied the minimum illumination ratio to the average illumination index.

Fourth, as shown in Figure 7, luminance values for 39 points were measured by considering the human viewing angle. The luminance of the light shelf reflector was also measured in order to obtain an average value by measuring eight points on the light shelf reflector. However, luminance could not be measured because measurement positions were not accessible when the light shelf angle was 0° and −10°, so luminance was measured on the window’s glass surface at the top of the light shelf. The luminance measurement location was set at 1.5 m from the floor and 5.5 m away from the skylight to consider the indoor space’s depth. The glare was analyzed based on the measured luminance and contrast values. The luminance measurement equipment is a handy type of high-precision luminance meter, the detailed specifications of which are shown in Table 6.

Fifth, the current study derived the process of natural light being reflected and flowing into a room depending on the light shelf angle and uses it as data to analyze the light shelf performance evaluation results. The natural light’s inflow process through the light shelf was visualized using AutoCAD (Autodesk Inc., San Rafael, CA, USA) by calculating the incidence angle and reflection angle. However, the scattered reflection was excluded from the visualization and processed as a kind of mirror reflection.

## 3. Result and Discussion

### 3.1. Performance Evaluation Results

The performance evaluation results for the three cases and environmental factors are presented in Table 7, Table 8, Table 9 and Table 10. 

First, as shown in Table 8, the lighting energy consumption maintained an appropriate indoor illumination of 500 lx for an external type light shelf with an angle of 0° and a reflectivity of 85% (Case 1) 5.290 kWh. On the other hand, the appropriate angles of a movable external light shelf with 85% reflectivity (Case 2) for energy saving were 30°, 30°, and 10° for the summer, middle season, and winter, respectively, and the energy consumption was 4.624 kWh. Meanwhile, the angle and reflectivity, only for energy saving, of the light shelf with a rolling reflector that can change its reflectivity (Case 3) were the same as an angle of 30° and a reflectivity of 97% for the summer, middle season and winter, and the lighting energy consumption was 4.288 kWh.

Second, the uniformity analysis depending on the light shelf angle is shown in Table 7 and Figure 8, and the details are as follows. In the summer, increasing the light shelf angle increased the amount of natural light that flows into a room through a light shelf reflection, so the indoor uniformity is improved. Even in the middle season, similar to the summer, the uniformity also improved as the light shelf angle increases. However, when the light shelf angle was 30°, as shown in Figure 9, the natural light that flowed through the light shelf reflection flowed directly onto the work surface without secondary reflection from the ceiling, so illuminance imbalance occurred, and the uniformity fell. Also, in the winter, when the light shelf angle was 20°, the uniformity dropped for the same reason as in the middle season with a light shelf angle of 30°. The angle of 30° was not suitable for improving uniformity because external natural light entered the lower part of the light shelf reflector.

Third, results of the luminance contrast depending on the light shelf angle, are reported in Table 7 and Figure 10. The light shelf’s angle control with a reflectivity of 85% had a luminance contrast value below 10, so glare was not generated in this condition. However, when the light shelf’s reflective surface angle was more than 10°, which is a light shelf angle that makes luminance measurement points exposed for measurement, its luminance contrast value was higher compared to the scenario in which the light shelf angle was –10° or 0°. Thus, it is advantageous not to expose the light shelf reflector to occupants by fixing the light shelf’s angle to 0° or less to decrease the luminance contrast value. Also, when the light shelf angle was 30° for the middle season and 20° for the winter, the external natural light could be exposed directly to the occupant through light shelf reflection. The luminance contrast value can also be high.

Fourth, as shown in Table 9, the increase of light shelf reflectivity increased the amount of natural light that flowed through light shelf reflection, which was advantageous for improving uniformity. However, as shown in Figure 11, it was unsuitable for improving the visual environment by increasing the luminance contrast. In particular, in the middle season, the condition of glare generation occurs when the light shelf angle was 30°. The reflectivity was 97%, so it gave an unpleasant visual environment to the occupants. Therefore, it should be advantageous to improve building energy saving and the indoor visual environment by appropriately changing the light shelf’s reflectivity rather than simply increasing it.

Given the information about uniformity and glare described above, the appropriate standard for light shelf 2 is shown in Table 11. It is different from the standard for a light shelf when only energy saving is considered. In particular, it is possible to improve lighting performance by appropriately changing reflectivity. It should also be effective for improving the visual environment related to uniformity and glare.

### 3.2. Discussion

The current study proposes a light shelf with a rolling reflector capable of changing reflectivity and verifies its performance. The results and additional suggestions are discussed in the following.

First, as shown in Figure 12, the light shelf using a rolling reflector that can change its reflectivity (Case 3) showed energy savings of 18.9% and 7.3%, respectively, over an external light shelf with an angle of 0° and a reflectivity of 85% (Case 1) and a movable external light shelf with a reflectivity of 85% (Case 2). Also, the light shelf with a rolling reflector that can change its reflectivity (Case 3) showed an energy savings of 13.6% and 5.7%, respectively, over the external light shelf with an angle of 0° and a reflectivity of 85% (Case 1) and the movable external light shelf with a reflectivity of 85% (Case 2), even when considering lighting energy saving, uniformity, and glare. This data proves the effectiveness of the lighting system proposed in the current study.

Second, changing the angle or reflectivity of a light shelf can cause a glare, creating an unpleasant visual environment for occupants. Thus, the following factors should be considered when designing a light shelf. When the light shelf angle is below 0°, the reflective surface is not exposed to occupants, resulting in a decreased luminance contrast value, and there is no glare problem. However, this causes a decrease in lighting performance. The movable light shelf with varied angles and a rolling reflector that can change its reflectivity (Case 3) can solve this problem.

Third, an external type of light shelf has the advantage of excellent performance [27]. However, its lighting performance deteriorates because of the decreased reflectivity caused by soiling. Dust accumulates on the reflector because of exposure to the outside. Therefore, maintenance is important for an external type light shelf, but there are also difficulties compared to an indoor type light shelf. As shown in Figure 13, the light shelf with a rolling reflector that can change its reflectivity, provides a space in the upper and lower sides of the light shelf that allows for installing additional cleaning brushes to partially remove soiling. This helps the light shelf maintain a certain level of lighting performance for a longer period. That is, the proposed light shelf with a rolling reflector improves lighting performance, uniformity, and glare, and keeps the light shelf reflector clean. This feature also highlights the effectiveness of the proposed system.

## 4. Conclusions

This study proposes a light shelf with a rolling reflector that can change its reflectivity to improve the lighting performance and visual environment and proves its effectiveness by conducting a performance verification. The conclusions are as follows.

First, the light shelf with a rolling reflector has three reflective films with different reflectivity connected and rolls the connected film using two axes. The design makes it possible to change the reflectivity of the light shelf. Also, maintenance of the external type light shelf will be easy because cleaning brushes can be installed on both sides of the light shelf. However, a light shelf with a rolling reflector that can change the reflectivity requires a system for rolling, so it has the disadvantage of increasing the shelf’s width and thickness.

Second, the increased light shelf angle increases the amount of natural light that flows deep into a room through the light shelf, so the indoor uniformity is improved. The increase in the reflectivity of a light shelf is advantageous in improving indoor uniformity. However, when the light shelf angle is 30° in the middle season, and the light shelf angle is 20° in the winter, natural light directly reaches occupants or the work surface through the light shelf’s reflection, which lowers uniformity. In addition, natural light does not reach the light shelf reflector in the winter, due to the solar altitude, if the light shelf angle is 30°. Instead, it partially blocks the natural light that can flow into the room, decreasing uniformity. This should be considered when improving the visual environment of an indoor space using a light shelf.

Third, the increase of the light shelf angle increases luminance contrast. In particular, when the light shelf angle is 30° in the middle season and is 20° in the winter, natural light directly reaches an occupant through the reflection of a light shelf, which is expected to create an unpleasant visual environment for the occupant. Also, the increased reflectivity increases luminance contrast. The light shelf angle of 30° and 98% reflectivity in the middle season causes a glare, but some can be removed by adjusting the reflectivity.

Fourth, it was found that the light shelf with a rolling reflector can reduce energy consumption by 18.9% and 7.3%, respectively, compared to a typical fixed light shelf and movable light shelf when only lighting energy saving is considered. Also, the light shelf with a rolling reflector that can change its reflectivity can reduce energy consumption by 13.6% and 5.7%, respectively, compared to a typical fixed light shelf and movable light shelf, even considering lighting energy saving, uniformity, and luminance. This proves the effectiveness of the light shelf with a rolling reflector.

This study verified the improvement of the visual environment according to the reflectivity and operation of a light shelf. The results will serve as a basis for future research to improve the performance of a light shelf, a kind of natural lighting system. However, as a preliminary investigation, there are some limitations to its extended use. The performance evaluation was performed using the south-facing direction only due to the mechanical limitations of the artificial sunlight. An evaluation that considers the azimuth angle of the sun and external illuminance per period, which may occur in an actual environment, should follow. Also, based on the current investigation, it is necessary to verify the applicability and effectiveness of the light shelf with a rolling reflector in terms of its installation cost, operation method, and various reflectivities in varied environmental conditions in the future studies.

## Figures and Tables

**Figure 1 ijerph-17-08338-f001:**
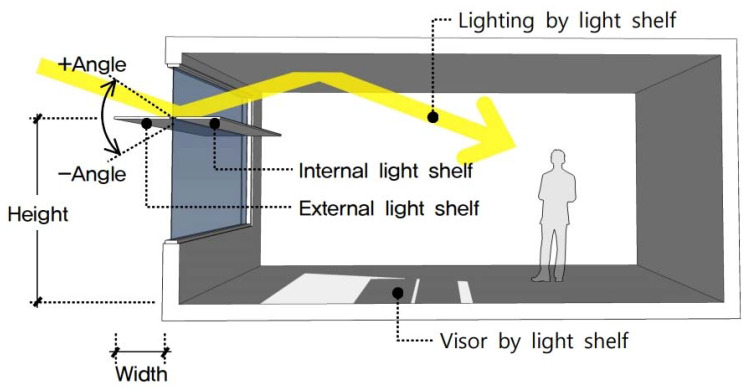
Light shelf concept and variables.

**Figure 2 ijerph-17-08338-f002:**
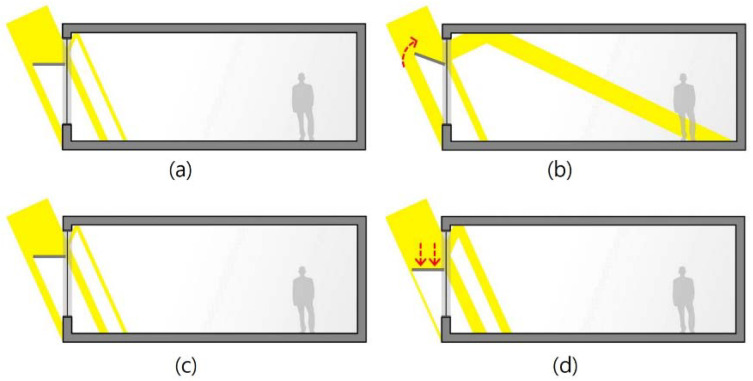
Natural light entering a room depending on light shelf angle and height: (**a**) natural light entering a room when light shelf angle is 0°; (**b**) natural light entering a room when light shelf angle is 20°; (**c**) natural light entering a room when light shelf height is 1.8 m; (**d**) natural light entering a room when light shelf height is 1.5 m.

**Figure 3 ijerph-17-08338-f003:**
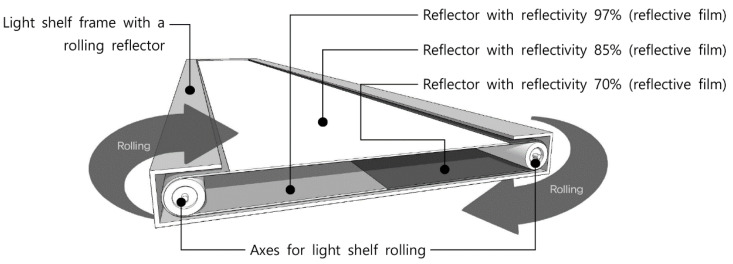
Internal structure of a light shelf with a rolling reflector.

**Figure 4 ijerph-17-08338-f004:**
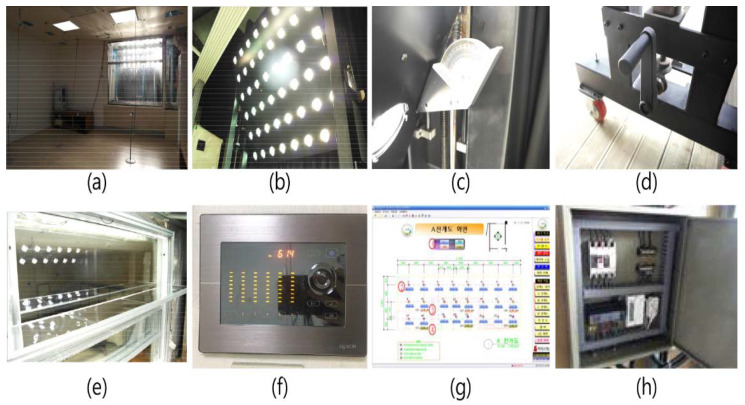
Overview of testbed and measurement apparatus: (**a**) performance evaluation testbed; (**b**) artificial solar irradiation apparatus; (**c**) control of light source angle of artificial solar irradiation apparatus; (**d**) lever to control light source altitude of artificial solar irradiation apparatus; (**e**) installed light shelf; (**f**) eight-level lighting controller; (**g**) illuminance sensor monitoring server; (**h**) energy monitoring system.

**Figure 5 ijerph-17-08338-f005:**
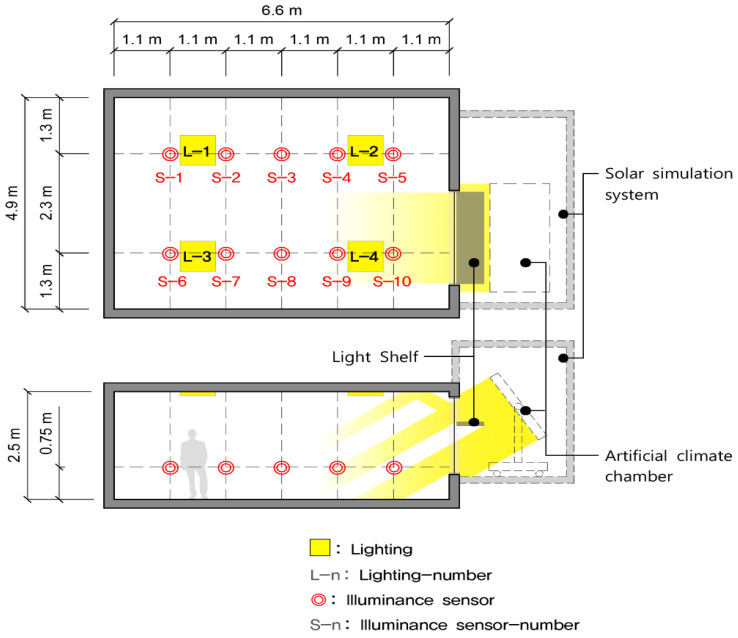
Cross-section and plane of testbed and sensor location.

**Figure 6 ijerph-17-08338-f006:**
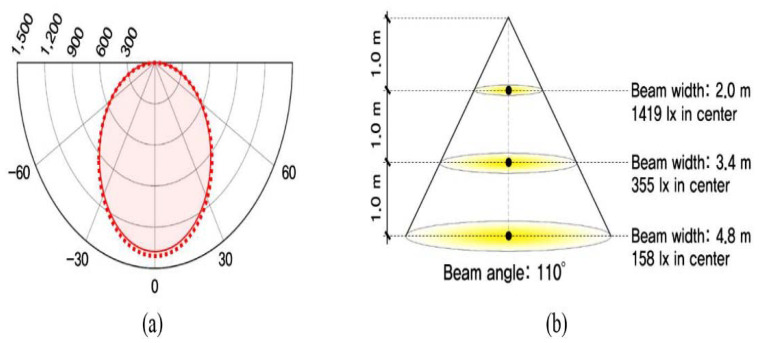
Light distribution curve and conical illuminance of lighting: (**a**) conical illuminance; (**b**) light distribution.

**Figure 7 ijerph-17-08338-f007:**
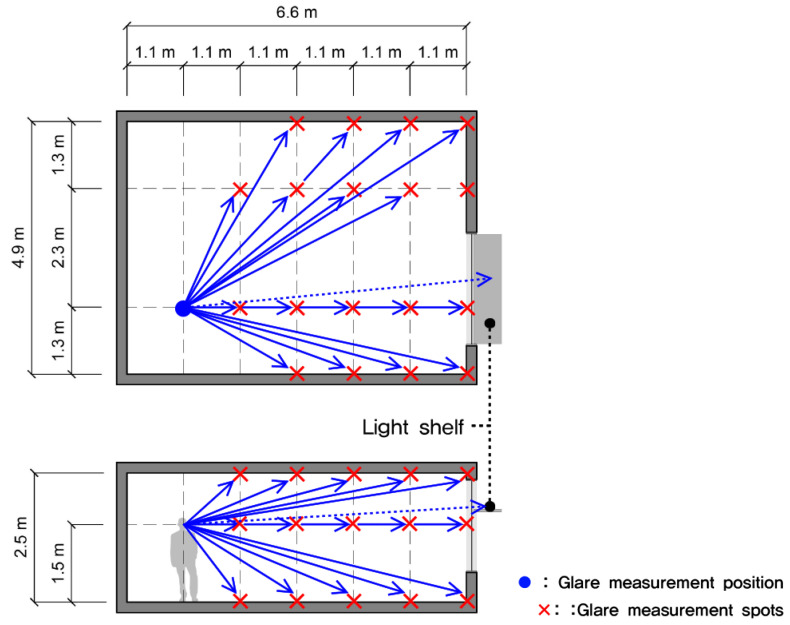
Glare measurement position and measurement spots.

**Figure 8 ijerph-17-08338-f008:**
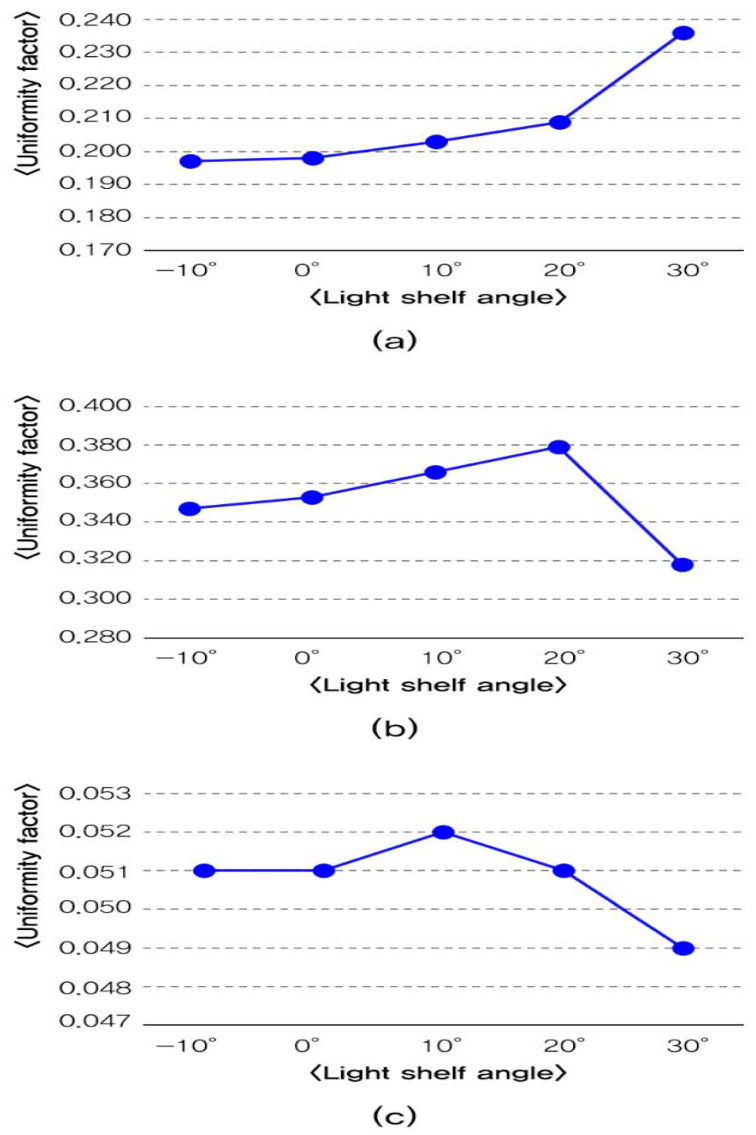
Analysis of uniformity per light shelf angle: (**a**) uniformity per light shelf angle in summer; (**b**) uniformity per light shelf angle in middle season; (**c**) uniformity per light shelf angle in winter.

**Figure 9 ijerph-17-08338-f009:**
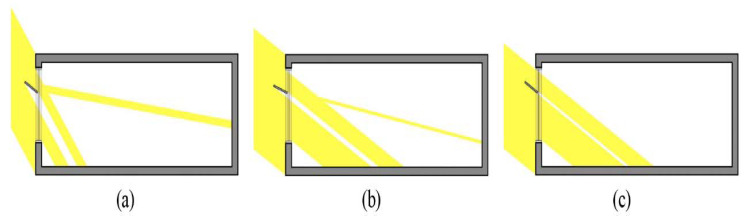
Diagram of light flow into a room per light shelf angle: (**a**) light shelf angle 30° in summer; (**b**) light shelf angle 20° in middle season; (**c**) light shelf angle 30° in winter.

**Figure 10 ijerph-17-08338-f010:**
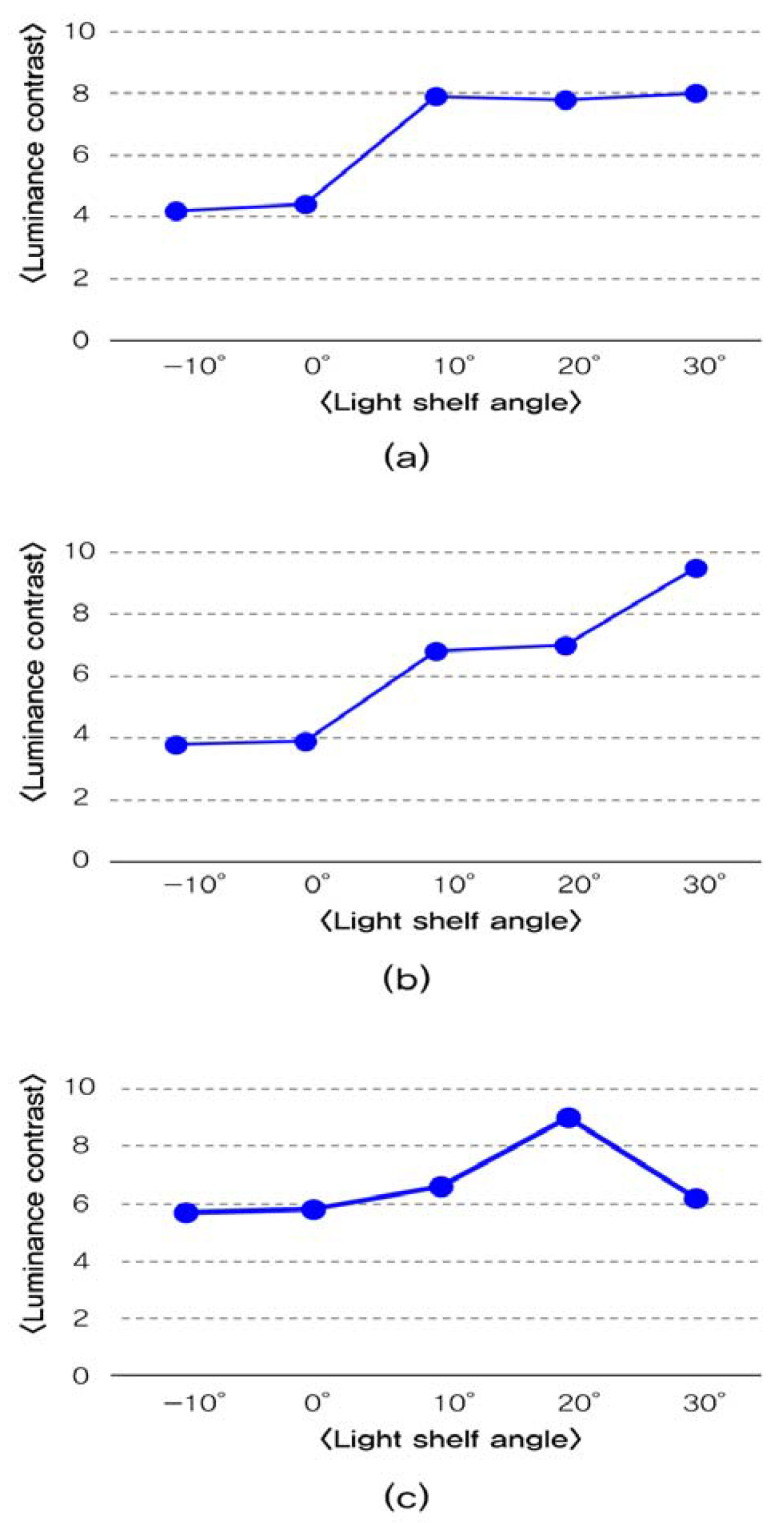
Analysis of luminance contrast per light shelf angle: (**a**) per light shelf angle in summer; (**b**) luminance contrast per light shelf angle in middle season; (**c**) luminance contrast per light shelf angle in winter.

**Figure 11 ijerph-17-08338-f011:**
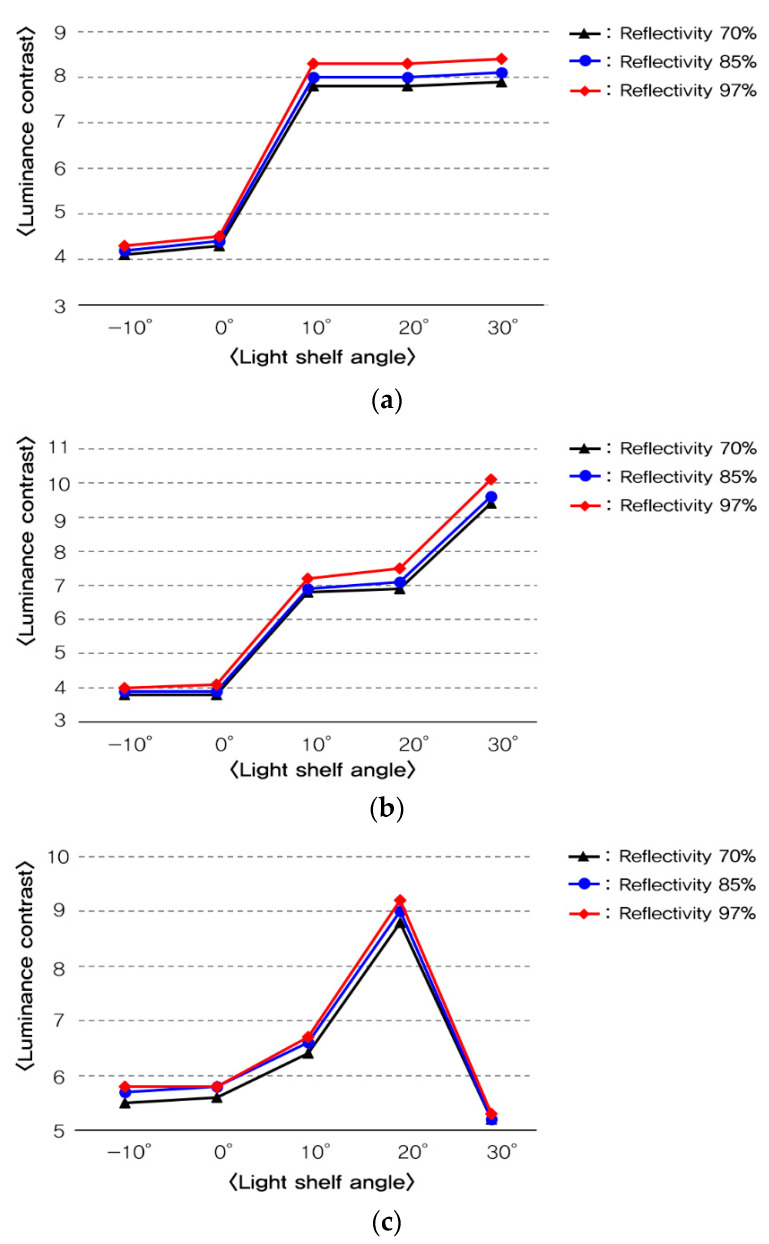
Analysis of luminance contrast per light shelf angle and reflectivity: (**a**) luminance contrast per light shelf angle and reflectivity in summer; (**b**) luminance contrast per light shelf angle and reflectivity in middle season; (**c**) luminance contrast per light shelf angle and reflectivity in winter.

**Figure 12 ijerph-17-08338-f012:**
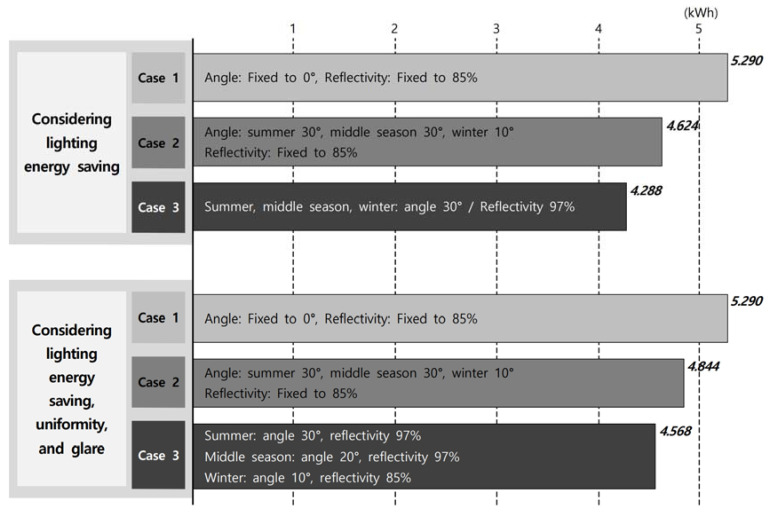
Performance evaluation results by case: total lighting energy usage.

**Figure 13 ijerph-17-08338-f013:**
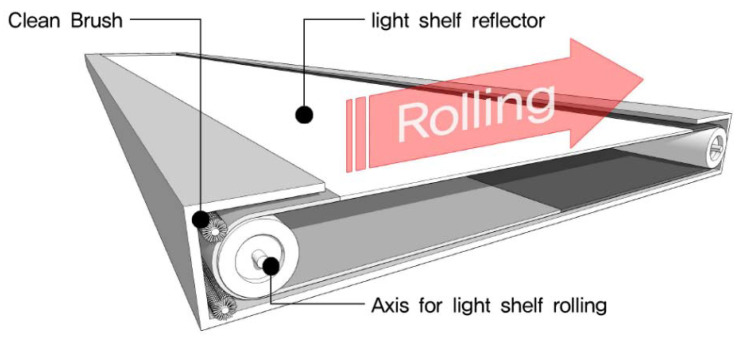
Maintenance of reflector through reflector rolling (application of clean brush system).

**Table 1 ijerph-17-08338-t001:** Review of prior studies on the light shelf.

Author	Light Shelf Variables to Improve Its Performance	Light Shelf Reflectivity Setup for Performance Evaluation	Consideration of Glare Caused by Light Reflectivity
Soler and Oteiza [13]	Fixed light shelf variables	Fixed to 91%	Not considered
Soler and Oteiza [14]	Fixed light shelf variables	Fixed to 91%	Not considered
Claros and Soler [15]	Fixed light shelf variables	Fixed to 84%	Not considered
Claros and Soler [16]	Width, angle, height, reflectivity	Performance evaluation was conducted by considering light shelf reflectivity (50%, 60%, 70%, 80%, 90%), but reflectivity was not changed per case during performance evaluation.	Not considered
Freewan et al. [17]	Fixed light shelf variables	Fixed to 85%	Not considered
Raphael [18]	Angle	Fixed to 60%	Not considered
Lim and Ahmad [19]	Height, shape	Fixed to 51.29%	Not considered
Lim and Heng [20]	Shape, height, width	Fixed to 51.29%	Not considered
Lee et al. [21]	Perforation rate of a reflector, angle, width	Fixed to 85%	Not considered
Berardi and Anaraki [22]	Fixed light shelf variables	Fixed to 80%	Not considered
Lee [23]	Angle	Fixed to 85%	Not considered
Lee et al. [24]	Curvature, angle	Fixed to 85%	Not considered
Meresi [25]	Width, angle, reflectivity	Performance evaluation was conducted by considering light shelf reflectivity (50%, 60%, 70%, 80%, 90%), but reflectivity was not changed per case during performance evaluation.	Not considered
Moazzeni and Ghiabaklou [26]	Width, angle	Fixed to 90%	Not considered
Mangkuto et al. [27]	Width, angle, reflectivity	Performance evaluation was conducted by considering light shelf reflectivity (60%, 70%, 80%, 90%), but reflectivity was not changed per case during performance evaluation.	Not considered

**Table 2 ijerph-17-08338-t002:** Consideration of optimal indoor illuminance standards by country.

Illuminance Standard (Country)	Type of Activity	Scope (lx)		
		Minimum Allowed Illumination	Standard Allowed Illumination	Maximum Allowed Illumination
IES (Illuminating Engineering Society) (USA) [34]	General (performance of visual tasks of medium contrast)	500	750	1000
JIS (Japan Industrial Standard) Z 9110 (Japan) [35]		300	500	600
KS (Korean Industrial Standards) A 3011 (Republic of Korea) [36]		300	400	600

**Table 3 ijerph-17-08338-t003:** Overview of testbed.

Room Size and Reflexibility	
Size	4.9 m (W) × 6.6 m (D) × 2.5 m (H)
Reflexibility	Ceiling 86%, Wall 46%, Floor 25%
Window size and material	
Size	1.9 m (W) × 1.7 m (H)
Type	Double glazed 12 mm 12 mm (3 Clean + 6 Air + 3 Clean)
Transmissivity	80%
Lighting	
Type	Eight-level dimming (LED-type) 4ea
Dimensions (mm)	600 × 600
Dimming range	10–100%
Energy consumption for phased light dimming	0 kWh (OFF), 12.3 kWh (Dimming Level 1), 18.3 kWh (Dimming Level 2), 22.0 kWh (Dimming Level 3), 27.7 kWh (Dimming Level 4), 34.0 kWh (Dimming Level 5), 38.5 kWh (Dimming Level 6), 42.6 kWh (Dimming Level 7), and 50.8 kWh (Dimming Level 8)
Illuminance sensor	
Sensing element	Silicon photosensor, with filter
Precision	±3%
Artificial solar light radiation apparatus	
Precision of solar light radiation	Grade A (according to ASTM E927-85)
Directions	South aspect
Energy monitoring system	
Model	SPM-141
Measurement capacity	Single-phase (220 V, 1–50 A)
Error rate	Within 2.0%

**Table 4 ijerph-17-08338-t004:** Configuration of meridian altitude of the sun and external illuminance for summer, middle season, and winter.

Season	Meridian Altitude	External Illuminance (lx)
Summer	76.5	80,000
Middle season	52.5	60,000
Winter	29.5	30,000

**Table 5 ijerph-17-08338-t005:** Cases for performance evaluation.

Case	Light Shelf Variables				
	Type	Height	Angle	Width	Reflectivity
1	External light shelf	1.8 m	Fixed to 0°	0.3 m	Fixed to 85%
2			−10°–30° (10° steps)	0.3 m	Fixed to 85%
3				0.38 m	Changeable to 70%, 85%, and 97%

**Table 6 ijerph-17-08338-t006:** Specifications and image of luminance meter.

	Details	Product and Measurement Images
Model	Luminance Meter LS-100 (Tokyo, Japan)	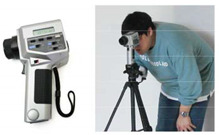
Type	SLR spot luminance meter for measuring light source and surface brightness	
Measuring range	FAST: 0.001–299,900 cd/m^2^, SLOW: 0.001–49,990 cd/m^2^	
Angle of view	9°	
Accuracy	±2% ±2 digits of displayed value	

**Table 7 ijerph-17-08338-t007:** Performance evaluation results of Case 1 and Case 2: analysis of uniformity, glare, and lighting energy consumption.

Season	Light Shelf		Illumination Sensor (lx)		U	Luminance (cd/m^3^)		L	Lighting Dimming Control: Light Number (Dimming Level)	Lighting Energy Consumption (kWh)
	A	R (%)	Min.	Ave.		Max.	Ave.			
Summer	−10	85	68.2	347.1	0.197	2848.4	681.5	4.2	1(8) + 3(8) + 2(5)	2.034
	0	85	73.6	372.3	0.198	3111.1	711.1	4.4	1(8) + 3(8) + 2(5)	2.034
	10	85	78.5	387.5	0.203	9224.7	1169.4	7.9	1(8) + 3(8) + 2(5)	2.034
	20	85	81.9	391.9	0.209	9294.4	1185.3	7.8	1(8) + 3(8) + 2(4)	1.939
	30	85	90.5	383.4	0.236	9814.2	1233.6	8.0	1(8) + 3(8) + 2(3)	1.855
Middle season	−10	85	120.2	346.8	0.347	4606.0	1202.6	3.8	1(8)→3(7)	2.802
	0	85	125.4	355.7	0.353	4918.0	1269.9	3.9	1(8)→3(6)	2.679
	10	85	131.5	359.1	0.366	12,539.7	1840.0	6.8	1(8)→3(5)	2.544
	20	85	139.9	369.1	0.379	12,588.6	1809.2	7.0	1(8)→3(5)	2.544
	30	85	149.6	469.7	0.318	17,659.0	1865.2	9.5	1(8)→3(4)	2.354
Winter	−10	85	249.5	4928.2	0.051	11,247.8	1959.7	5.7	1(5)	0.510
	0	85	255.9	5044.6	0.051	11,472.6	1989.3	5.8	1(5)	0.510
	10	85	265.8	5142.5	0.052	12,843.6	1947.7	6.6	1(4)	0.415
	20	85	270.5	5278.8	0.051	18,432.5	2048.3	9.0	1(4)	0.415
	30	85	247.9	5087.7	0.049	12,202.4	1952.7	6.2	1(5)	0.510

Note; A: Angle, R: Reflectivity, Min.: Minimum Ave.: Average, U: Uniformity factor, L: Luminance contrast.

**Table 8 ijerph-17-08338-t008:** Light shelf angle for lighting energy saving and total lighting energy consumption for Case 1 and Case 2.

Case	Season	Light Shelf Angle for Energy Saving	Lighting Energy Consumption (kWh)	Total Lighting Energy Consumption (kWh)
1	Summer	Fixed to 0°	2.034	5.290
	Middle season		2.679	
	Winter		0.577	
2	Summer	30°	1.855	4.624
	Middle season	30°	2.354	
	Winter	10°	0.415	

**Table 9 ijerph-17-08338-t009:** Performance evaluation results of Case 3: analysis of uniformity, glare, and lighting energy consumption.

Season	Light Shelf		Illumination Sensor (lx)		U	Luminance (cd/m^3^)		L	Lighting Dimming Control: Light Number (Dimming Level)	Lighting Energy Consumption (kWh)
	A	R(%)	Min.	Ave.		Max.	Ave.			
Summer	−10	70	54.3	280.3	0.194	2273.3	548.1	4.1	1(8) + 3(8) + 2(6)	2.102
		85	67.4	341.8	0.197	2879.7	678.1	4.2	1(8) + 3(8) + 2(5)	2.034
		97	76.1	384.0	0.198	3235.4	745.9	4.3	1(8) + 3(8) + 2(5)	2.034
	0	70	58.6	300.6	0.195	2482.9	571.8	4.3	1(8) + 3(8) + 2(6)	2.102
		85	72.7	366.6	0.198	3145.3	707.5	4.4	1(8) + 3(8) + 2(5)	2.034
		97	82.2	411.9	0.200	3533.8	778.3	4.5	1(8) + 3(8) + 2(4)	1.939
	10	70	62.7	313.0	0.200	7362.2	940.5	7.8	1(8) + 3(8) + 2(5)	2.034
		85	77.8	381.7	0.204	9326.2	1163.6	8.0	1(8) + 3(8) + 2(4)	1.939
		97	88.1	428.9	0.205	10642.8	1280.0	8.3	1(8) + 3(8) + 2(3)	1.855
	20	70	65.3	316.6	0.206	7417.8	953.2	7.8	1(8) + 3(8) + 2(5)	2.034
		85	81.0	386.1	0.210	9396.6	1179.4	8.0	1(8) + 3(8) + 2(4)	1.939
		97	91.8	433.8	0.212	10723.2	1297.3	8.3	1(8) + 3(8) + 2(3)	1.855
	30	70	71.9	309.6	0.232	7832.7	992.0	7.9	1(8) + 3(8) + 2(5)	2.034
		85	89.3	377.6	0.236	9922.2	1227.4	8.1	1(8) + 3(8) + 2(3)	1.855
		97	101.3	424.2	0.239	11323.0	1350.1	8.4	1(8) + 3(8) + 2(2)	1.799
Middle season	−10	70	96.0	280.1	0.343	3676.1	967.1	3.8	1(8) + 3(8) + 2(1)	3.419
		85	118.4	341.6	0.347	4656.7	1196.6	3.9	1(8) + 3(7)	2.802
		97	133.7	383.8	0.348	5314.1	1316.3	4.0	1(8) + 3(5)	2.544
	0	70	100.6	286.3	0.351	3925.0	1021.3	3.8	1(8) + 3(8)	3.049
		85	124.1	349.1	0.355	4972.1	1263.6	3.9	1(8) + 3(6)	2.679
		97	140.2	392.2	0.357	5674.0	1390.0	4.1	1(8) + 3(4)	2.354
	10	70	104.2	289.5	0.360	10007.8	1479.7	6.8	1(8) + 3(8)	3.049
		85	129.5	353.1	0.367	12677.6	1830.8	6.9	1(8) + 3(5)	2.544
		97	147.8	396.7	0.373	14467.4	2013.9	7.2	1(8) + 3(4)	2.354
	20	70	111.2	297.9	0.373	10046.9	1455.0	6.9	1(8) + 3(7)	2.802
		85	138.2	363.3	0.380	12727.1	1800.2	7.1	1(8) + 3(4)	2.354
		97	149.6	391.1	0.382	14923.9	1980.2	7.5	1(8) + 3(4)	2.354
	30	70	118.9	378.9	0.314	14093.5	1500.0	9.4	1(8) + 3(7)	2.802
		85	147.6	462.1	0.319	17853.2	1855.9	9.6	1(8) + 3(4)	2.354
		97	159.9	498.5	0.321	20673.7	2041.5	10.1	1(8) + 3(2)	2.074
Winter	−10	70	202.2	3962.6	0.051	8865.9	1600.0	5.5	1(5)	0.510
		85	247.3	4853.3	0.051	11247.8	1959.7	5.7	1(4)	0.415
		97	280.8	5481.4	0.051	12769.6	2213.3	5.8	1(4)	0.415
	0	70	207.4	4052.1	0.051	9070.1	1624.2	5.6	1(5)	0.510
		85	253.6	4962.9	0.051	11472.6	1989.3	5.8	1(4)	0.415
		97	287.9	5605.2	0.051	13078.8	2251.4	5.8	1(4)	0.415
	10	70	215.3	4130.9	0.052	10154.0	1590.2	6.4	1(5)	0.510
		85	263.3	5059.4	0.052	12843.6	1947.7	6.6	1(4)	0.415
		97	298.9	5714.1	0.052	14732.4	2204.3	6.7	1(4)	0.415
	20	70	219.1	4246.1	0.052	14746.0	1672.4	8.8	1(5)	0.510
		85	268.0	5200.6	0.052	18432.5	2048.3	9.0	1(4)	0.415
		97	304.3	5873.6	0.052	21251.6	2313.4	9.2	1(4)	0.415
	30	70	200.9	4092.5	0.049	10045.0	1913.6	5.2	1(5)	0.510
		85	245.7	5012.4	0.049	10250.02	1952.7	5.2	1(4)	0.415
		97	278.9	5661.1	0.049	10506.3	1991.8	5.3	1(4)	0.415

Note; A: Angle, R: Reflectivity, Min.: Minimum Ave.: Average, U: Uniformity factor, L: Luminance contrast.

**Table 10 ijerph-17-08338-t010:** Light shelf angle for lighting energy saving and total lighting energy consumption for Case 3.

Season	Light Shelf Variables for Energy Saving		Lighting Energy Consumption (kWh)	Total Lighting Energy Consumption (kWh)
	Angle	Reflectivity		
Summer	30	97	1.799	4.288
Middle season	30	97	2.074	
Winter	30	97	0.415	

**Table 11 ijerph-17-08338-t011:** Analysis of light shelf variables and lighting energy consumption to improve energy saving and visual environment.

Case	Season	Light Shelf Variables to Improve Energy Saving, Uniformity, and Glare		Lighting Energy Consumption (kWh)	Total Lighting Energy Consumption (kWh)
		Angle	Reflectivity		
2	Summer	30	Fixed to 85%	1.885	4.844
	Middle season	20		2.544	
	Winter	10		0.415	
3	Summer	30	97%	1.799	4.568
	Middle season	20	97%	2.354	
	Winter	10	85%	0.415	

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
