# Peer review of "A Basic Study on the Performance Evaluation of a Movable Light Shelf with a Rolling Reflector That Can Change Reflectivity to Improve the Visual Environment"

_ijerph, 2020, doi:10.3390/ijerph17228338_

Round 1

Reviewer 1 Report

The subject of the article Development of a Movable Light Shelf with a Rolling Reflector that can Change Reflectivity to Improve the Visual Environment interest at this time. The article is well structured, although there are some aspects that need to be improved and some corrections and/or observations that the authors should consider in order to improve the article:

The introduction has to be completed with up-to-date scientific literature, which in turn will give you a richer up-to-date reference on the subject. Authors should clarify these aspects.

Line 58-70 Reference is made to the fact that other studies with the same purpose do not take into account the variation of reflectivity and glare. Do these studies refer to the cost of the installation and its relationship to potential energy savings?

Line 49-55. It is not clear how the system proposed in this study can be adapted in the windows. Each window has different sizes. It is necessary to explain how it can be placed. Should a different device be made for each window? Do we also have to produce new windows for each window in each room in order to adapt this system? No reference to the possible cost of installing the system in each window   

Line 109-110. It should be explained how it can be controlled if it exceeds much of the 500 lx and the glare practically on the part of people who are in the room in a real way. Is it all automatic?

Line 111-112. Reference is made to two engines so they must have electrical connection. Has it been taken into account that if the engines are on the outside they have to be stable in time resisting rain and temperature differences?

-Line 122-125. It is necessary to explain in a clearer way how the reflectivity is activated or changed when a person enters the room. Can the person who enters decide?

Line 192-218. Each room in a situation has different sizes and a different window size. Explain whether there should be an LED light system with a computer program that regulates the intensity of light emitted by each LED luminaire depending on the luxes in each part of the room. If so, what would be the cost of lighting dimming control and illumination sensor? What could be the cost of the software that takes into account the different variables?

The conclusions point out that the application of some of the ideas drawn from the study be implemented with economic cost of all the elements necessary for installation to be able to compare with the expected energy savings after installation.

Author Response

According to the reviewers’ comments, the manuscript underwent minor revisions. All the changes are marked in the resubmitted manuscript. We elaborate on how we did this, and how we respond to the reviewer’s comments in a seprate file attached.

Reviewer 2 Report

The study presents a movable light shelf with a rolling reflector that can change reflectivity to improve the visual environment of humans. The tackled problem is worth investigating, namely due to its direct impact on human visual comfort and lighting energy consumption. 

The paper is well organized and divided into clear sections, the methods are clearly explained and the results are well discussed. The conclusion summarizes the main outcomes of the research and the author highlights the limitations of the proposed technology. However, the manuscript requires some amendments before accepted for publication. The comments are as follow:

  1. The quality of Figures 8, 10, 11, and 12 must be improved, the figures are barely readable.   
  2. Lines 192-218: I recommend to summarize the procedure using a flow chart.
  3. The English language is very good, however, there is a lot of avoidable repetitions in the text, for instance, (this study is repeated more than 40 times in the manuscript) and several other terms, I recommend to revise and improve this aspect.

Author Response

(The authors gave the same response as above.)

Reviewer 3 Report

Dear Authors,

The work presented deal with a relevant topic considering the present status of development of studies considering the improvement of the energy efficiency of buildings

The objective of the paper is clearly presented, as well as the methodology used in its development. Most of the possible drawbacks of the testing installation are presented and argued.

A new proposal of shelf is presented introducing the capability of changing the refection rate available along with other used variables in this type of systems as the angle position.

Many data are presented as results of the experiment developed, and a final discussion and the conclusion chapter summarize the optimal configuration found according to the testing context used and the three most significant lighting parameters (energy efficiency, Em, U and glare).

Some minor issues that I would recommend to review:

Figure 4 presents many different elements in such a small size that they give a Little amount of useful information. They do not clarify much the testing installation.

The resolution of several figures, mostly number 12 is low and the text included in not readable and it includes a good summary of the conclusions obtained.

Even though the English used is fine to understand the content of the work a review of the text can be suggested to improve it. Some bad use of prepositions as in line 147 are presented (i.e. LINE 147 based on related studies [26] FOR Seoul, Korea). Moreover, is several sentences of the text the same words are repeated several times too close: “with a rolling reflector that can change the reflector’s reflectivity” -> “with a rolling reflector that can change its reflectivity”

Chapter 2.2 include a very large paragraph with much information that is hard to follow. I would probably recommend making some divisions.

height of working area is usually taken as: 0.85 m. Even though there it is possible to guess that there is not significant different from 0.75 m.

Table 3 present some typing mistakes:

“4.9 m (w) × 6.6 m (D) × 2.5 m (H)”   “w” should be a capital letter

It seems to be missing a horizontal line below “Window size and material” and “Energy monitoring system”. Also the size of the room does not seem to need its line.

Also, there are some small incongruences between the text and the table:

The different equipment used are not described in and homogeneous way. I.E. only the model of the Energy monitoring system is given.

Some Data given in this table are irrelevant for the work of the paper.

Text: “support 9-step dimming control” Table: “Eight-level dimming (LED-type)” 4ea”

Text: “including lighting OFF,” Table: “Dimming range 10%–100%”

The element “4ea” is never detailed

Table 7 also have some of them.

I could be useful to indicate which dimming level (1-8) refers to nominal or minimum power of the LED luminaires. ¿Does ‘0’ or none means that the luminaire is OFF?

How did you choose the nominal power of the luminaires? It is significant to me from table 7 & 9 that either in summer and in the mid seasons the Em is much lower than 500 lx. Also the light emission pattern seems to be too narrow as for using only 4 luminaires? A night/dark simulation/measurement have been done?

I appreciate the effort to considerate the “maintenance factor” related to the shelf proposed. However, considering that it is a movable system placed outdoors, the analysis made that just using brushes they will keeps the light shelf reflector clean and within reflective specifications (97%) is simplistic and most probably not functional in urban ambient.

The conclusions should make a reference of the indicated limitation of the study declared in chapter 2.2: “Its azimuth angle was limited to the south direction.”

Sincerely

Author Response

(The authors gave the same response as above.)

Reviewer 4 Report

This article deal with a movable light shelf with a rolling reflector that can change the reflectivity. the system’s ability had been analyzing in terms of energy and the visual environment: reflectivity, visual environment and glare. The results showed that he light shelf system proposed in this study achieved an energy savings of 13.6% and 5.7%. Additionally, the light can reduce energy consumption by 13.6% and 5.7%, respectively, compared to a typical fixed light shelf and movable light shelf, even considering lighting energy saving, uniformity, and luminance.

The study has potential to be a useful contribution to the journal of Int. J. Environ. Res. Public Health. The topic of save energy and improve visual environment is meaningful and relevant for an international audience.

However, the manuscript presently brief lack of clarification to present a clear and significant scientific contribution.

The image resolution of all figures needs to be improved. Further, the energy expenditure estimation of the light shelf with a rolling reflector with varied reflectivity must be indicated.

Line 14 The phrase “produce basic data.” ... needs to be improved for better understanding and to avoid confusion.

Line 44 Include bibliography related to model of glare evaluated.

Line 101 Some more data from other countries can be incorporated to provide a more global analysis and perspective. Correct bold type in “IES (USA) [33]”.

Line 142: Indicate the exact model of the lighting system used: Include specific data (model) of lamp.

Line 349-350 “Also, maintenance of the external type light shelf will be easy because cleaning brushes can be installed on both sides of the light shelf.” It is necessary to justify that it is easy or change, the cleaning of the system can be a disadvantage in relation to energy efficiency. Is it airtight? If so, could there be reflections losses? Can dirt cause scratches? In what proportion does it affect performance? Clarify

Line 387. Replace authors with author.

.

Author Response

(The authors gave the same response as above.)
